# Are Aortic Root and Ascending Aorta Diameters Measured by the Pediatric versus the Adult American Society of Echocardiography Guidelines Interchangeable?

**DOI:** 10.3390/jcm10225290

**Published:** 2021-11-14

**Authors:** Maria Luz Servato, Gisela Teixidó-Turá, Anna Sabate-Rotes, Laura Galian-Gay, Laura Gutiérrez, Filipa Valente, Ruben Fernandez-Galera, Guillem Casas, Angela López-Sainz, M. Teresa González-Alujas, Augusto Sao-Aviles, Ignacio Ferreira, Jose Rodríguez-Palomares, Arturo Evangelista

**Affiliations:** 1Department of Cardiology, University Hospital Vall d’Hebron, CIBERCV, 08035 Barcelona, Spain; luzservato@gmail.com (M.L.S.); gisela.tt@gmail.com (G.T.-T.); lauragaliangay@gmail.com (L.G.-G.); lauraguga@gmail.com (L.G.); filipaxaviervalente@gmail.com (F.V.); rubenfdezgalera@gmail.com (R.F.-G.); gcasasmasnou@gmail.com (G.C.); kelals@hotmail.com (A.L.-S.); teresagonzalu@gmail.com (M.T.G.-A.); saoavilesaugusto@gmail.com (A.S.-A.); nachoferreira@secardiologia.es (I.F.); jfrodriguezpalomares@gmail.com (J.R.-P.); 2Department of Pediatric Cardiology, University Hospital Vall d’Hebron, CIBERCV, 08035 Barcelona, Spain; annasabaterotes@gmail.com; 3Teknon Medical Center-Quirón Salud, Heart Institute, 08022 Barcelona, Spain

**Keywords:** aorta, aortic dimensions, echocardiography, guideline’s recommendations

## Abstract

Ascending aorta diameters have important clinical value in the diagnosis, follow-up, and surgical indication of many aortic diseases. However, there is no uniformity among experts regarding ascending aorta diameter quantification by echocardiography. The aim of this study was to compare maximum aortic root and ascending aorta diameters determined by the diastolic leading edge (DLE) and the systolic inner edge (SIE) conventions in adult and pediatric patients with inherited cardiovascular diseases. Transthoracic echocardiograms were performed in 328 consecutive patients (260 adults and 68 children). Aorta diameters were measured twice at the root and ascending aorta by the DLE convention following the 2015 American Society of Echocardiography (ASE) adult guidelines and the SIE convention following the 2010 ASE pediatric guidelines. Comparison of the diameters measured by the two conventions in the overall population showed a non-significant underestimation of the diameter measured by the SIE convention at root level of 0.28 mm (CI −1.36; 1.93) and at tubular ascending aorta level of 0.17 mm (CI −1.69; 2.03). Intraobserver and interobserver variability were excellent. Maximum aorta diameter measured by the leading edge convention in end-diastole and the inner edge convention in mid-systole had similar values to a mild non-significant underestimation of the inner-to-inner method that permits them to be interchangeable when used in clinical practice.

## 1. Introduction

Maximum aortic root and ascending aorta diameters have important clinical value in the diagnosis, follow-up, and surgical indication of many aortic diseases. Therefore, accurate and standardized reproducible measurement techniques are required [1,2]. Transthoracic echocardiography (TTE) is the most widely used imaging technique in clinical practice given its availability, accuracy, and low cost. However, there is no consensus among experts regarding the method used to take ascending aorta measurements by echocardiography. Although the American Society of Echocardiography (ASE) guidelines recommended that two-dimensional echocardiographic measurements be performed in the parasternal long-axis view, adult guidelines in 2015 recommended this measurement at end-diastole by the leading-to-leading edge convention (DLE) [3], while the pediatric guidelines recommended the inner-to-inner edge convention in mid-systole (SIE) [4]. The inner-to-inner edge (I-I) convention by echocardiography was also supported by the 2010 American College of Cardiology and American Heart Association guidelines [5].

The real value differences that the DLE and SIE methods may generate in maximum aorta diameter measurement are not well defined and might have clinical management implications, particularly in adolescents in the transition from pediatric to adult departments. Although several publications reported normal values using both conventions in healthy adult subjects [6,7,8], the impact of both measurement methods in patients with cardiovascular diseases has not been evaluated.

The aim of this study was to assess the differences in maximum aortic root and ascending aorta diameters measured by TTE using both ASE guidelines recommendations in adults and pediatric patients.

## 2. Materials and Methods

Three hundred and twenty-eight consecutive patients (260 adults and 68 children) with a diagnosis of Marfan, Loeys–Dietz and Ehlers–Danlos Syndrome, familial aorthopathies, bicuspid aortic valve, hypertrophic cardiomyopathy, and arrhythmogenic diseases followed at the inherited cardiovascular diseases Unit of the Vall d’Hebron Hospital were included in the study. Patients underwent an echocardiographic study in the pediatric or adult echocardiography laboratories depending on their age at the annual follow-up control. Adult patients were considered to be >18 years of age. Exclusion criteria were previous aorta surgery, atrial fibrillation, and a poor echocardiographic window. The study was approved by the local ethics committee (Ethical Committee of Clinical Studies of Vall d’Hebron Hospital; ethical approval code 4975).

### 2.1. Echocardiography

Standardized echocardiographic examinations were performed with Vivid E95, Vivid E9 and Vivid 7-GE Healthcare equipment. All images were digitally acquired and analyzed off-line using Echo PAC software (GE Healthcare). Images were acquired during breath-hold in the parasternal long-axis view using a multifrequency ultrasound transducer. Measurements of aorta diameters were taken perpendicular to the long axis of the aorta, at the level of the sinuses of Valsalva, and ascending aorta 1 mm above the sinotubular junction. Measurements were taken off-line by a single expert echocardiographer following the recommendations of the 2015 adult ASE guidelines at end-diastole, using the leading edge to leading edge (L-L) convention (DLE). After a one-week interval, a second blind measurement was obtained by the same expert following the 2010 pediatric ASE guidelines by the maximum aorta diameter measured from the inner edge to inner edge of the aortic wall at mid-systole (SIE) (Figure 1).

The value of the diameter in each case was obtained from the mean of the diameters measured in three consecutive beats.

To assess intraobserver variability, aorta diameter was measured in 20 randomly selected patients by a single echocardiographer with a one-week interval, and interobserver variability was assessed in a further 20 randomly selected patients by two experienced echocardiographers (M.L.S. and A.E.). Readers were blinded to each other and were allowed to select the cardiac cycles to be measured, but the average of three consecutive beats was recorded.

### 2.2. Statistical Analysis

In the demographic table, quantitative variables were described as mean and standard deviation (SD) or median and interquartile range while the qualitative variables were presented as percentages. For quantitative variables, the differences were calculated with the T-test or Mann–Whitney test as appropriate or according to the distribution. Fisher’s exact test or Chi-square were used for qualitative variables, as appropriate. The measurements of each aortic segment were compared to determine the differences between diameters obtained by both methods. This analysis was carried out using Bland–Altman plots, mean absolute differences, and intraclass correlation coefficients (ICC). An ICC ≥ 0.8 indicated an excellent correlation. *p* values < 0.05 indicated statistically significant differences. Intraobserver and interobserver variabilities were calculated with Lin’s correlation coefficient and reported as the mean of the absolute differences and ICC.

## 3. Results

### 3.1. Demographic and Clinical Characteristics

Demographic data are shown in Table 1. Briefly, the adult group included 260 patients (55.4% men; median age: 55.2, range: 32–75-years), and the pediatric group included 68 patients (66.2% boys; median age: 10.9, range: 6–14-years).

### 3.2. Median Aortic Diameters and Mean of the Differences Using Both Techniques

Mean aortic root diameters obtained by DLE and SIE were 34.16 ± 6.78 mm and 33.88 ± 6.90 mm, respectively (*p* = 0.600), while at ascending aorta level they were 33.20 ± 7.75 mm and 33.02 ± 7.53 mm, respectively (*p* = 0.763), as shown in Table 2. Comparison of the diameters measured by the two conventions in the overall population showed a non-significant underestimation of the diameter measured by the SIE convention at root level of 0.28 mm (CI −1.36; 1.93) and 0.17 mm (CI −1.69; 2.03) at the tubular ascending aorta level. Scatterplots describing the correlation between the DLE and SIE techniques and Bland–Altman analyses at each aorta segment in the total population are shown in Figure 2A,B.

In the adult group, the mean underestimation of maximum diameter by SIE compared with DLE at the root level was 0.27 mm (CI −1.15; 1.70) and 0.33 mm (CI −1.20; 1.88) at the ascending aorta level (Figure 2C,D), while in the pediatric group it was 0.30 mm (CI −1.99; 2.61) and 0.47 mm (CI −2.87; 1.93), respectively (Figure 2E,F) (Table 2).

### 3.3. Intraobserver and Interobserver Variability

Intraobserver variability was good by DLE (ICC 0.97–95% CI 0.95–0.99) and SIE (0.89 95% CI 0.82–0.97) at the level of the aortic root and ascending aorta (ICC 0.94–95% CI 0.89–0.99) and (ICC 0.87–95% CI 0.77–0.96), respectively. Interobserver variability was also good by DLE (ICC 0.97–95% CI 0.95–0.99) and SIE (ICC 0.93–95% CI 0.88–0.98) at the level of the aortic root and ascending aorta (ICC 0.81–95% CI (0.66–0.96)) and (ICC 0.94–95% CI (0.90–0.99)), respectively. (Table 3).

## 4. Discussion

The present study showed maximum aorta diameter measured following pediatric [4] and adult ASE [3] guidelines recommendations to be similar with a mild non-significant underestimation of the SIE method compared with DLE, which allows them to be interchangeable when used in clinical practice. Furthermore, an excellent correlation was found between diameters measured by both conventions and reproducibility was equally high for both techniques.

With the significant advances in multimodality cardiac imaging, a number of techniques are currently used for noninvasive assessment of aortic disease [9]. AHA Guidelines in 2010 recommended that external aorta diameter be reported for computed tomography (CT)- or cardiac magnetic resonance (CMR)-derived size measurements but, in contrast, internal diameter for echocardiography [5]. These recommendations generated significant controversy in aorta measurements. Several echocardiographic studies reported their results of measuring aortic root and ascending aorta diameters by the I-I edge in diastole [10] in an attempt to be consistent with other vascular imaging modalities (CT and CMR) and increase the intermodality comparison of aorta measurements [11,12,13]. In addition, normal aortic root values were reported by this I-I edge convention measured in end-diastole [6,7,13,14]. Nevertheless, a significant underestimation of echocardiographic diameters compared with CT or CMR was observed when the measurement was taken by the SIE convention [15].

However, we showed that I-I edge has similar diameter values to DLE if it is measured in meso-systole. In line with our results, Muraru et al. [14] reported no significant differences between SIE and DLE diameters in a series of 218 healthy adult volunteers. Systolic expansion of aorta diameter measured by the leading edge convention was 2 mm, similar to the mean aortic wall thickness value of 2.4 ± 0.8 mm. Diameters obtained by both conventions such as DLE, with inclusion of anterior aortic wall thickness, and SIE, with systolic expansion of aorta size, have similar diameter values. In a large cohort with 1687 healthy adult subjects, end-diastolic measurements correlated strongly with mid-systolic measurements in men and in women for all aortic root diameters [16] Bossone et al. [7] compared aortic measurements using the DLE and SIE conventions in a large cohort of healthy adult individuals and reported normal values by SIE. In that study, the maximum diameter at the level of the sinuses of Valsalva was significantly greater by the DLE method (0.21 ± 1.35 mm; *p* < 0.001). Conversely, the ascending aorta diameter was greater using the SIE (0.26 ± 0.98 mm; *p* < 0.001 [7]. Similarly, Son et al. [8] in a smaller series of 112 healthy Korean volunteers reported that DLE aortic diameters were larger than SIE diameters at the sinus of Valsalva but smaller at the level of the ascending aorta. Those results might indicate that systolic expansion could be greater in the ascending aorta than in the aortic root. Our study analyzed these diameters in patients with or without aorta dilation in the adult and pediatric populations and found no significant differences using DLE and SIE conventions. However, SIE diameter tended to be larger in pediatric patients which may be related to higher aortic distensibility.

In clinical practice, aorta diameters measurements by CT and CMR are taken in diastole since, in this phase of the cardiac cycle, the dimensions of the aorta are more stable and less influenced by blood pressure changes, cardiac output, or heart rate. Thus, a similar argument may be applied for TTE. Furthermore, the DLE measurement convention links to a large body of historical and prognostic data that have long guided clinical decision-making. Thus, to optimize the concordance among measurements with different imaging techniques and reproducibility, we consider that the aorta should be measured at the same moment of the cardiac cycle, such as in end-diastole.

One study published by our group [17] showed that the measurements taken on TTE using the DLE convention yielded the best agreement with those taken on CT and CMR using the I-I convention in end-diastole. By contrast, use of the I-I convention in end-diastole on TTE significantly underestimated the dimensions obtained with CT and MRI. These results were confirmed by other authors [15,18]. Most studies comparing the various methods used to assess aortic dimensions with 2D TTE in healthy individuals showed that the L-L convention yielded larger dimensions than the I-I convention [6,7,8,11]. Several authors justify the use of the L-L convention by TTE owing to the worse axial resolution of harmonic 2D-TTE compared to CT and CMR. Harmonic imaging improves the interphase definition of the structures but reduces the axial resolution of the ultrasound pulse and may overestimate aortic wall thickness by 20%, resulting in a systematic underestimation of the inner–inner diameter [17,19]. In our study, anterior aorta wall thickness was 3.2 ± 1.6 mm by TTE, 1.4 ± 1.2 mm by MRI and only 0.8 ± 0.7 mm by CT. However, differences between L-L and I-I are due not only to spatial echo resolution but also to structures included in the measurement, such as the anterior aorta wall itself. Therefore, aorta diameter measurement by 2D TTE may change by >5 mm with the inclusion or exclusion of the aorta walls, with the L-L convention being the best echocardiographic approach to reduce this limitation [17,20]. From a clinical point of view, underestimation by the I-I convention in diastole may adversely affect patient prognosis by delaying surgery when the cutoff values are validated using the L-L convention.

Pediatricians classically measure the aorta by TTE using the SIE convention. These different recommendations seem very difficult to unify and can generate confusion, particularly in adolescents making the transition of care from pediatric to adult cardiology services. However, the present study shows differences in stable hemodynamic conditions to be very small and without significant clinical implications.

Limitations. The main limitations of the study were the small sample size, especially in the pediatric population, and lack of blood pressure measurement and analysis of demographic data such as BMI at the time of the echocardiographic study. However, the study does provide relevant information compared to previous studies by including pediatric and adult patients with cardiovascular diseases and a larger range of values. Aortic root and ascending aorta diameter differences by DLE vs. SIE conventions in this population had not been previously analyzed.

## 5. Conclusions

Maximum aorta diameter measured by the leading edge convention in end-diastole and inner edge convention in mid-systole has similar values with a mild non-significant underestimation of the inner-inner method, which allows them to be interchangeable when used in clinical practice. Furthermore, the correlation and reproducibility of aorta diameters measured by both conventions are excellent.

## Figures and Tables

**Figure 1 jcm-10-05290-f001:**
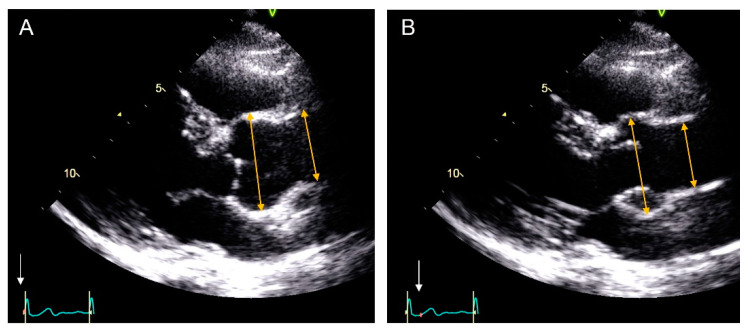
Measurement of maximum aorta diameters by echocardiographic images in parasternal long-axis view. Aortic root and ascending aorta measuring technique as recommended by the ASE guidelines for adults (leading-to-leading edge in diastole) (**A**) and by the ASE guidelines for children (inner-to-inner edge in systole) (**B**). The white arrows mark the measurement time-points in relation with the cardiac cycle oh the electrocardiogram (ECG).

**Figure 2 jcm-10-05290-f002:**
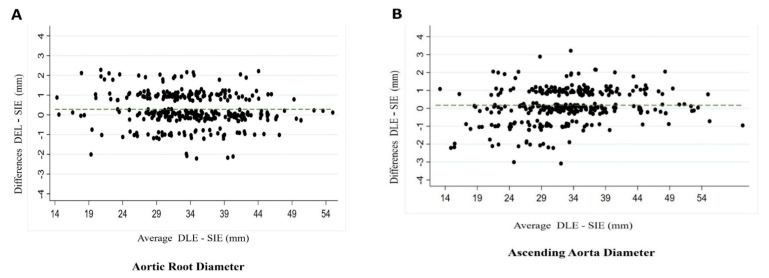
Bland–Altman plots showing the mean of the differences at the level of the aortic root (**A**) and ascending aorta (**B**) in the total population. Bland–Altman plots showing the mean of the differences at the level of the aortic root (**C**) and ascending aorta (**D**) in the adult population. Bland–Altman plots showing the mean of the differences at the level of the aortic root (**E**) and ascending aorta (**F**) in the children population.

**Table 1 jcm-10-05290-t001:** Clinical characteristics for all subjects (*n* = 328).

Parameter		Adults (*n* = 260)	Pediatrics (*n* = 68)	Total Population (*n* = 328)	*p*
Age (year)		65.18 (51.78;75.26)	10.88 (5.92;14.12)	59.86 (27.88;72.21)	<0.001
Male		144 (55.4%)	45 (66.2%)	189 (57.6%)	0.11
Diagnosis of Inherited Diseases	Marfan Syndrome	67 (25.76%)	31 (45.59%)	98 (29.88%)	
	Loeys-Dietz/ Ehler-Danlos Syndrome	19 (7.31%)	4 (5.88%)	23 (7.01%)	
	Familial aortopathy	21 (8.08%)	5 (7.35%)	26 (7.93%)	
	Bicuspid aortic valve	88 (33.85%)	28 (41.18%)	116 (35.36%)	
	Hypertrophic cardiomyopathy	49 (18.85%)	0 (0%)	49 (14.94%)	
	Arrhythmogenic Diseases	16 (6.15%)	0 (0%)	16 (4.89%)	

**Table 2 jcm-10-05290-t002:** Median aortic diameters and mean of the differences in adult, children, and total population.

	DEL (mm)	SIE (mm)	*p*	Mean of the Differences (mm)
Adult population (*n* = 260)				
Aortic root	35.83 (5.84)	35.51 (5.90)	0.534	0.27 (CI −1.15; 1.70)
Ascending aorta	35.62 (6.42)	35.23 (6.36)	0.486	0.33 (CI −1.20; 1.88)
Children population (*n* = 68)				
Aortic root	27.99 (6.51)	27.61 (6.78)	0.739	0.30 (CI −1.99; 2.61)
Ascending aorta	24.09 (4.95)	24.52 (5.04)	0.600	0.47 (CI −2.87; 1.93)
Total population (*n* = 328)				
Aortic root	34.16 (6.78)	33.88 (6.90)	0.600	0.28 (CI −1.36; 1.93)
Ascending aorta	33.20 (7.75)	33.02 (7.53)	0.763	0.17 (CI −1.69; 2.03)

**Table 3 jcm-10-05290-t003:** Intraobserver and interobserver variability in 20 random patients.

		Adult Guidelines (DLE)			Pediatric Guidelines (SIE)	
Aortic root	ICC	*p*	95% CI	ICC	*p*	95% CI
Interobserver variability	0.97	<0.001	0.95−0.99	0.93	<0.001	0.88−0.98
Intraobserver variability	0.97	<0.001	0.95−0.99	0.89	<0.001	0.82−0.97
Ascending aorta	ICC	*p*	95% CI	ICC	*p*	95% CI
Interobserver variability	0.81	<0.001	0.66−0.96	0.94	<0.001	0.90−0.99
Intraobserver variability	0.94	<0.001	0.89−0.99	0.87	<0.001	0.77−0.96

ICC = intraclass correlation coefficient.

## Data Availability

The data presented in this study are available on request from the corresponding author.

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
