# Peer review of "Are Aortic Root and Ascending Aorta Diameters Measured by the Pediatric versus the Adult American Society of Echocardiography Guidelines Interchangeable?"

_jcm, 2021, doi:10.3390/jcm10225290_

Round 1

Reviewer 1 Report

In the paper ‘Are aortic root and ascending aorta diameters measured by the paediatric versus the adult American Society of echocardiography guidelines interchangeable?’ the authors report an observational study in which they compared measurements of the aortic root and ascending aorta in diastole leading edge to leading edge (L-L) and in systole inner tot inner edge (I-I). The study population consisted of 328 patients with a inherited cardiovascular condition (aortopathy or cardiomyopathy). Their main finding is that there is a non-significant underestimation of the diameters when the aorta is measured in systole I-I both at the level of the root and the ascending aorta.

This a very nice and useful study which addresses a common question in clinical practice. It’s well and clearly written but the content may still be improved in several ways.

  1. The method used to measure ascending aorta has not been explained properly. The authors describe ‘around 5mm above ST-junction’. Could the authors be more precise on what ‘around’ means and could they explain why were the measures not taken at the level of the right pulmonary artery? On Fig1. measurement seem to have been taken at a level too close to the aortic root. An additional comment on Fig1.: it could be meaningful to add ECG on the images to be able to evaluate the exact moment at the cardiac cycle when measurements have been performed.
  2. The authors included both patients with and without aortopathy. Aortic characteristics in the latter group might differ from the 1st and lead to different results depending on the underlying cardiovascular disease. Did the authors analyse the results in patients with and without aortopathies? In the same line, judging from the Bland-Altman plots there are patients were higher difference in measurement (between 2.5-3.5mm) are recorded. Could the authors explain what characteristics did these patients have? If relevant this information should be added to the text.
  3. As the authors themselves describe in the discussion, their results are different to previous studies. Do the authors have any thought on why this could be?

Some additional minor comments:

  1. Please specify in the abstract that this study has been performed in patients with an inherited cardiac condition.

Please add to line 56 which cardiovascular conditions have been included

Author Response

  1. Thank you very much for this observation. We made a typo error since we measured ascending aorta by parasternal long-axis view 1 mm above ST-junction. This is the method used in our previous article (Rodríguez-Palomares JF et al. Multimodality Assessment of Ascending Aortic Diameters: Comparison of Different Measurement Methods. J Am Soc Echocardiogr. 2016 Sep;29(9):819-826.e4) and in other articles (Muraru D et al. Ascending aorta diameters measured by echocardiography using both leading edge-to-leading edge and inner edge-to-inner edge conventions in healthy volunteers. Eur Heart J Cardiovasc Imaging. 2014 Apr;15(4):415-22; Saura D et al. Two-dimensional transthoracic echocardiographic normal reference ranges for proximal aorta dimensions: results from the EACVI NORRE study. Eur Heart J Cardiovasc Imaging. 2017 Feb;18(2):167-179). We have rewritten the phrase on page 2, line 72: “at the level of the sinuses of Valsalva, and ascending aorta 1 mm above the sinotubular junction”

  2. We agree that this important information is missing. We have included the ECG in Figure 1 and indicated with a white arrow the moment of the cycle where it was measured (A: tele-diastole, B: meso-systole). We have attached the new Figure 1 in the manuscript on page 3, line 89. We have added the comment: “The white arrows mark the measurement time-points in relation to the cardiac cycle on the electrocardiogram (ECG)” on page 3, lines 93-94.

  3. This is a good observation. As we stated in reply 1, the aim of the study was to analyze the differences in the maximum root and ascending aorta diameters using both conventions. In more than 30% of patients with genetic or bicuspid aortopathy, the aortic diameter was normal, and patients with other genetic diseases had slight aortic dilation. These data did not influence the results of the study since the inclusion of a wide spectrum of diameters was more important than the etiology of the aortic dilation.

  4. This is a very interesting point. We believe that this result is because we have included children, which may have higher aortic distensibility and aorta systolic expansion than adult patients.
  5. Some additional minor comments:

    1. We agree with your suggestion. We have added it on page 1, lines 18-19: The aim of this study was to compare maximum aortic root and ascending aorta diameters determined by the diastolic leading-edge (DLE) and the systolic inner-edge (SIE) conventions in adult and pediatric patients with inherited cardiovascular diseases”.

    2. We agree with this comment. We have added this information on page 2, lines 57-58: “…with a diagnosis of Marfan, Loeys-Dietz and Ehlers-Danlos Syndrome, familial aorthopathies, bicuspid aortic valve, hypertrophic cardiomyopathy, and arrhythmogenic diseases…”

Reviewer 2 Report

I enjoyed reading the manuscript "are aortic root and ascending aorta diameters measured by the pediatric versus the adult American Society of echocardiography guidelines interchangeable". It's a clinically important issue since serial longitudinal follow-up of patients with aortic pathology is essentially done by transthoracic echocardiography and especially in the transition period from pediatric to adult cardiology data measured by different methods could be misleading if not potentially dangerous. 

The authors discuss literature comparing both methods studied to gold standard CT or CMR measurements and although SIE could slightly underestimate aortic diameters, data from their study showed that differences were very small and not clinically significant. It seems to me that this is an important finding in order to reassure patients about the values of different methodologies since pediatricians and adult guidelines remain different. It could be very useful (and probably most centers do this) to compare individual echo measurements with CT or CMR measurements during transition. The only problem seems the ascending aorta, which is no surprise of course, in which the mean of differences seems bigger especially in the younger population. Sudden echocardiographic changes in this area, whether measured with the same or with different methods, warrant control by CT or CMR.

Maybe the authors could add another recent publication: 

Transthoracic echocardiographic reference values of the aortic root: results from the Hamburg City Health Study

Author Response

Thank you very much for your comments. This is a very interesting recent publication and we added it in the references and discussion.